# Managers' Perceptions of Telework in Relation to Work Environment and Performance

Tea Korkeakunnas [1,*], Marina Heiden [1], Malin Lohela-Karlsson [1,2,3] and Komalsingh Rambaree [4]

1. Centre for Musculoskeletal Research, Department of Occupational Health Sciences and Psychology, Faculty of Health and Occupational Studies, University of Gävle, SE-801 76 Gävle, Sweden
2. Department of Public Health and Caring Sciences—Health Services Research, Uppsala University, SE-751 22 Uppsala, Sweden
3. Centre for Clinical Research, Region Västmanland—Uppsala University, Hospital of Västmanland, SE-721 89 Vasteras, Sweden
4. Department of Social Work and Criminology, Faculty of Health and Occupational Studies, University of Gävle, SE-801 76 Gävle, Sweden
* Correspondence: tea.korkeakunnas@hig.se; Tel.: +46-26-64-5057

**Abstract:** The study aimed to investigate managers' perceptions of telework in relation to the work environment and the performance of their organizations. It was grounded on interviews with 17 managers from public and private organizations in Sweden using a phenomenographic research approach. The managers had experience in leading employees who teleworked, and their contact information was provided by the organizations. The results showed that managers perceived that telework led to more focus on delivery at work. When telework enabled working undisturbed and getting more work done, employees started to question the need to come to the workplace. Further, some employees changed their behaviors at the workplace; they tended to close their office doors to avoid distractions. Others valued social activities when working at the workplace. Managers also reported that telework could make small close-working teams even closer, but they could lose contact with others outside the teams. In addition, telework led to quicker but less informed decisions, which may benefit productivity in the short term but not in the long term. The findings of the study highlight managers' perspectives on the consequences of telework in an organization, which is important for maintaining organizational sustainability.

**Keywords:** remote work; telecommuting; manager; wellbeing; effectivity; deliver; work routines; social community; added value; attractivity

## 1. Introduction

Telework, which is defined as a work arrangement in which employees are not located in a central office building, but rather work at a distant location—for example, from home or from a summer cottage [1]—has been practiced by several global organizations over the years. During the COVID-19 pandemic, many employees and employers with little or no previous experience of telework had to practice telework, leading to a transformation of the physical and psychosocial work environment [2,3], with potential consequences for both individuals and organizations.

During the pandemic (2019–2022), 50–60% of employees in Nordic countries teleworked, as did 37% of all employees in the EU [4]. Preliminary findings from recent studies show that one-third of employees worldwide and two out of three employees in Sweden would like to work from home to some extent [2,5], indicating that employees expect telework to be a part of modern work life. Consequently, organizations need to find ways of practicing telework that benefit the employees as well as the organization.

Previous research has investigated the effects of telework on individual outcomes such as job satisfaction, work-life balance, conflict, wellbeing, and happiness [6,7]. On the

one hand, telework can increase employees' work satisfaction and motivation and enhance their health and work–life balance, leading to higher productivity [8–10]. On the other hand, working from home without physical contact with other work colleagues can lead to feelings of social isolation and perceived threats to professional advancement [8,10]. Organizations may expect better productivity, better employee health, and greater wellbeing when the implementation of telework is planned thoughtfully and intentionally [7,11].

Managers need to consider individual needs as well as the organizational consequences of telework. From an employer's perspective, telework can be seen as a risk for an organization due to the loss of control and reduced coordination [12]. Furthermore, managers have concerns about following employees' behavior and job performance when the latter are teleworking [12,13]. Previous studies have shown that trust is of the greatest importance for managers' attitudes towards telework and, thus, for telework to be adopted in an organization [7,14,15]. Other factors that affect telework permissions are the type of work performed, the need for interaction with colleagues, and support from top management in terms of, e.g., telework policies [7]. In fact, even if the organizational culture is not supportive of telework, management may be inclined to allow it if telework policies exist, and management is highly dependent on their team's performance [16]. If managers are supportive and have positive relationships with their employees, they can improve leadership, trust, and social exchange relationships, which are essential factors when leading teleworkers [7,17].

In an organizational context, telework can be understood in relation to neo-institutional theory. Organizations are continuously subjected to pressures from outside and inside the organization, and these pressures form the attitude toward telework [18]. Within an organization, a factor of particular importance for adopting telework is the nature of the work tasks being performed. If the work tasks can be performed remotely, employees may be allowed to telework if they can be trusted to do their job regardless of where they are located or if their performance can be easily assessed. The prerequisites are related to the organizational culture. Daniels (2021) postulated that beliefs in the importance of personal flexibility for work performance likely increase the implementation of telework in an organization, whereas beliefs in the importance of bureaucratic control and stability likely do the opposite [18].

The implementation of telework in an organization may affect individual employee behaviors depending on how the employees perceive that telework can be practiced, e.g., some employees may find it useful for withdrawing from interactions with colleagues, whereas others may feel obligated to be reachable all the time. Telework may also affect the culture and norms, e.g., which amount of telework is considered acceptable in an organization. Both aspects can influence the attitude toward telework in the organization. Overall, neo-institutional theory is found to be suitable for discussing and contrasting the findings from this study. To understand the impact of telework in organizations, studies are needed to elucidate how the work environment and performance of each organization are affected. More specifically, information about organizational aspects such as decision making, how performance is evaluated, and work communication can shed light on how important areas benefit from or are challenged by the implementation of telework in an organization. Knowing what telework implies for management is critical to ensure that any future more permanent modifications to telework policies benefit both employees and the organizations and provide organizational sustainability. Hence, this study aimed to investigate how managers perceived telework arrangements in relation to the work environment and the performance of their organizations.

## 2. Methods

### 2.1. Study Design

The present study used a phenomenographic research approach. The goal of phenomenography, in this context, is to discover the qualitatively different ways in which people experience, conceptualize, realize, and understand various aspects of telework

in the world around them [19]. The data collection took the form of a semi-structured interview, and the questions were constructed to allow participants to reflect on their own experiences [19]. Phenomenography inspired the authors to make interview questions. Moreover, previous studies influenced the set of questions.

### 2.2. Participants

This study was conducted within the 6-year Forte program "Flexible work—opportunity and challenge" [20]. Contact persons from different organizations (part of the Forte program) were asked to provide a list of names and contact information for managers with experience in leading employees who teleworked. The list needed to contain men and women of different ages and tenures.

In total, 17 managers (53% men and 47% women) were included in the study. They were strategically selected to represent male and female managers of different ages and tenures to increase the transferability of the findings (Table 1). Their ages ranged between 37 and 64 years, and they worked in public and private organizations. Most of them did not have any experience in managing teleworkers before the COVID-19 pandemic. At the time of the interview, their employees teleworked approximately 2–3 days per week, and the managers did so themselves 1–2 days per week, but there were some exceptions (i.e., some employees and managers teleworked for more than 90% of their working time). Most of the organizations had a policy that their employees and managers were able to telework for 49% of their working time, and this decision was made by the highest management. All managers lived in Sweden, and they had been working for two months to over 20 years in the same position at their respective organizations. Except for two interviewees who were part of the top management, the interviewees were middle managers.

**Table 1.** Information about the participants.

| Name | Age | Sex | Management Level | Experience as a Manager | Number of Employees | Sector | Type of Organization |
|---|---|---|---|---|---|---|---|
| 1 | 42 | Man | Middle Manager | 1 year | 13 | Production | Private |
| 2 | 39 | Man | Middle Manager | 3, 5 years | 16 | Production | Private |
| 3 | 40 | Man | Middle Manager | 20 years | 10 | Production | Private |
| 4 | 64 | Woman | Top manager | 16 years | 230 | Municipality | Public |
| 5 | 52 | Man | Top manager | 10 years | 150 | Municipality-Culture, and leisure | Public |
| 6 | 46 | Woman | Middle Manager | 7 years | 25 | Municipality-Supply chain | Public |
| 7 | 42 | Woman | Middle Manager | 1, 5 years | 21 | Municipality-Environment | Public |
| 8 | 51 | Man | Middle Manager | 12 years | 13 | IT | Public |
| 9 | 37 | Man | Middle Manager | 2, 5 years | 10 | Municipality-Economy | Public |
| 10 | 55 | Woman | Middle Manager | 12 years | 15 | Municipality-Economy | Public |
| 11 | 52 | Man | Middle Manager | 7 years | 15 | Municipality-IT | Public |
| 12 | 43 | Woman | Middle Manager | 12 years | 18 | Municipality | Public |
| 13 | 53 | Man | Middle Manager | 20 years | 21 | Municipality-Communication | Public |
| 14 | 60 | Woman | Middle Manager | 9 years | 6 | Municipality | Public |
| 15 | 55 | Man | Middle Manager | 27 years | 14 | Municipality-Food | Public |
| 16 | 33 | Woman | Middle Manager | 5 years | 20 | Municipality-Properties | Public |
| 17 | 50 | Woman | Middle Manager | 8 years | 11 | Municipality-Energy | Public |

### 2.3. Data Collection

Initially, three pilot interviews were made with two women and one man (aged above 45 years and with more than 5 years of working experience in public organizations) to see if the questions were suitable for this study. After that, small changes were made to the

interview guide. The final interview guide was formed, and it contained 27 open-ended questions about the background information of the managers, organizations, the managers' working teams, how employees worked, decision making, the work environment, telework, and the possibilities and challenges of telework. For detailed information on the interview guide, see Supplementary File S1. All interviewees were encouraged to speak openly about their experiences. Follow-up questions such as "Could you elaborate on that?" were used to clarify their answers. They were asked to reflect on the same themes and questions, but the follow-up questions differed depending on the interview. One of the authors (T.K.) performed the interviews. The atmosphere during the interviews was relaxed.

The interviews were held between February 2022 and August 2022. Participants received information about the study and the consent form by e-mail. Before the interviews started, participants gave their written consent to participate in the study. The interviews were conducted individually and held online by using the Microsoft Teams® platform (version 1.6.00.4472). They were recorded with the permission of the participants. The interviewing stopped at 17 interviews when no new information had come up from the last three interviews and saturation was reached [21,22]. The interviews lasted between 35 and 70 min. This study was approved by the Swedish Ethical Review Authority (2019-06220).

### 2.4. Analyses

ATLAS.ti Win (version 9.1.6.0) was used to analyze the data [23]. All of the interviews were listened to and transcribed. The transcriptions were moved to ATLAS.ti Win (version 9.1.6.0) [22], where quotations, codes, sub-categories, and categories—relationship-based aspects, task-based aspects, and added value (see Figure 1)—were built. The categories were selected based on groundedness (i.e., number of times of occurrence of a theme), density (i.e., number of linkages that a theme had with other themes), differences between the interviews, and what was new in the research field. All authors participated in reading the transcripts and analyzing the gathered data to ensure that no relevant data were inadvertently excluded or that irrelevant data were included, thereby increasing the credibility of the findings. Categories and sub-categories were identified through a consensus among the authors. A category is a collection of similar data sorted into the same place, and it can be compared and contrasted with other categories [24]. The following steps of phenomenographic analyses were followed [25].

- Familiarization: transcriptions were read several times to get to know the material.
- The most representative statements and quotations for the study were selected (e.g., "Our check-ins have strengthened and integrated us as a team").
- Grouping: The statements were sorted by similarities (creating different codes, such as better performance), e.g., "Things that took maybe two hours at the office—you can do now in 45 min . . . ", "Better productivity when you can choose your working environment".
- Comparison: Different statements were compared to identify sources of variations (e.g., "Some employees have chosen to close their doors. They have started to do so or to sit at home so that they can work in peace" and "Some employees say: could we have meetings at the office?"). In addition, the essence of the similarities was classified (creating different sub-categories, e.g., work routines).
- After sub-categories, three categories were created: relationship-based aspects, task-based aspects, and added value.

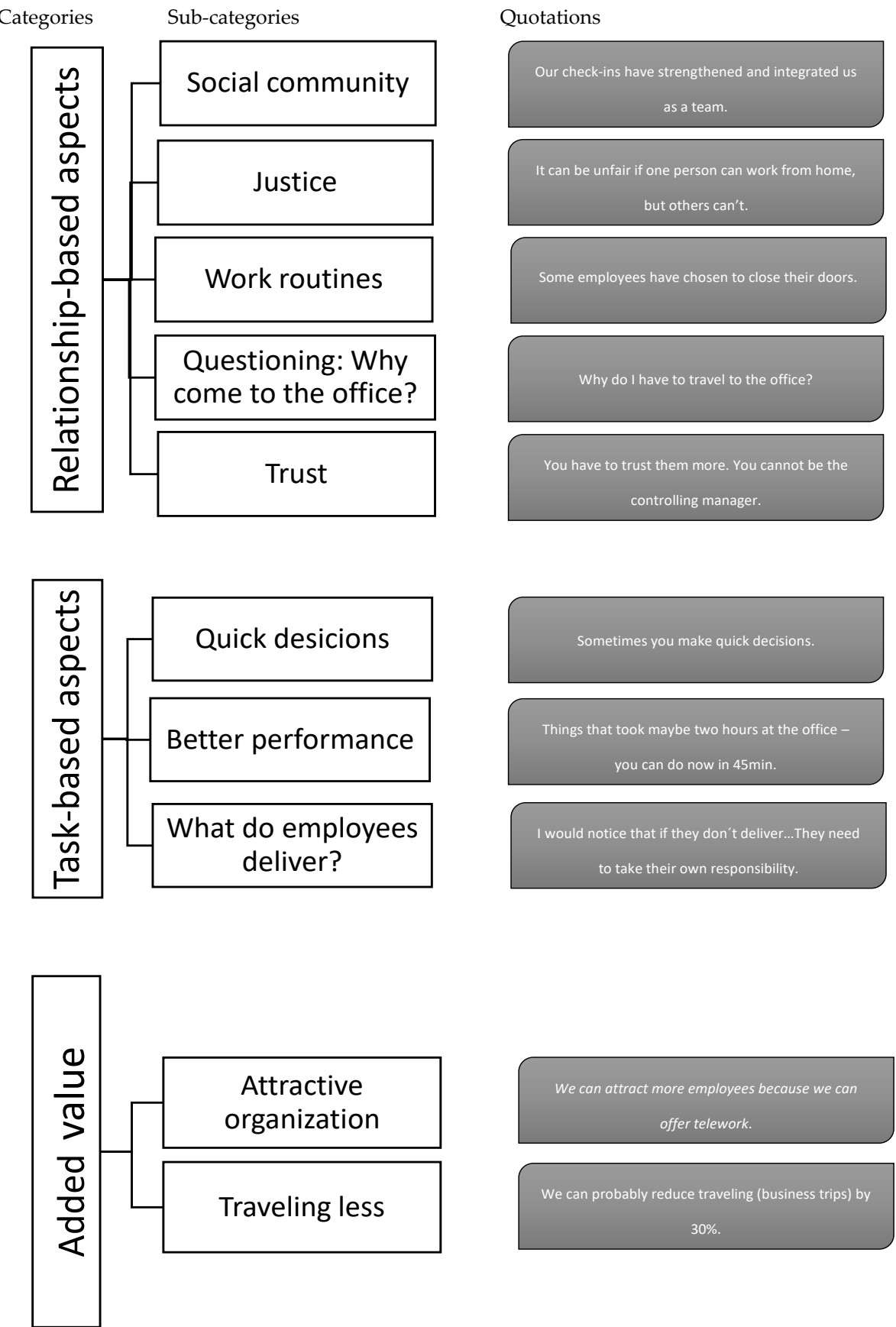

**Figure 1.** Categories, sub-categories, and quotations. Figure 1 provides some examples of quotations.

## 3. Results

The analysis resulted in three categories: relationship-based aspects, task-based aspects, and added value (Figure 1). Variations were found in some sub-categories.

### 3.1. Relationship-Based Aspects

#### 3.1.1. Social Community

According to the interview responses, telework affects the social community at work. Several managers reported that their teams had become closer during the pandemic and that they had better team cohesion due to the regular morning meetings initiated during the pandemic. In particular, if the team was small (approximately 10 people), employees had become more familiar with each other. Number 12 from a public organization described the situation in her team:

> *Our check-ins have strengthened and integrated us as a team. We have never known so much about each other's work as we do now.*

However, some managers thought that telework had not affected their working teams, but it had affected the social community in their organizations. In particular, if there were some newcomers in the organization, the employees did not recognize new employees from other departments. Number 12 from a public organization said:

> *My team has lost contact with other groups in the organization. In the team, there is no difference, but in the department, you lose a lot—you lose overhearing (work-related questions) which happens during the coffee breaks.*

Therefore, a relatively large number of managers thought that physical meetings were needed to build relationships. It was also more suitable to speak about personal matters face-to-face according to the interviewees. Number 16 from a public organization said:

> *You must have a mix of meetings...You speak differently when you meet physically. It can also be challenging to have serious or difficult discussions online (if somebody's relative has passed away).*

In the specific cases in which managers teleworked for almost all of their working time and many of their work colleagues were located in different cities or countries, but they knew each other well, regular face-to-face meetings were not needed because social connection, commitment, and cohesion were built online. Number 3 from a private organization said:

> *It is easier to meet online. We have gotten used to telework.*

#### 3.1.2. Justice

Telework can create problems between different working groups and professions in the organization when some employees have the opportunity to telework and some do not. In some branches, it can be difficult to telework due to work assignments—not all work assignments are suitable for telework, such as in work in a production department at a factory. Number 3 from a private organization described the situation:

> *It can be unfair if one person can work from home, but others can't—I think It can create more of a "we and they" feeling.*

This can lead to irritation between departments in the organization. When some employees can telework and others cannot, it can lead to an unpleasant atmosphere in the organization and create challenges in cooperations among different working groups according to the managers. Number 16 from a public organization said:

> *Telework can create moaning among the employees in the organization.*

#### 3.1.3. Work Routines

When (part-time) telework is allowed, it leads to new work routines and behaviors at the office due to telework. Employees do not want to be distracted at the office—they want

to have their own space and work in peace more than before. Number 13 from a public organization described the situation:

> *Employees react more than before at the office. Some employees have chosen to close their doors. They have started to do so or to sit at home so that they can work in peace.*

However, some employees and managers want to keep more traditional work routines—they want to meet each other, participate in small talk, and have coffee breaks every now and then when they are working at the office. They appreciate the social atmosphere at the office. Number 2 from a private organization said:

> *Some employees say: could we have meetings at the office?*

### 3.1.4. Questioning—Why Come to the Office?

Introducing telework may also lead to difficulties in getting employees to come to the office, especially, if employees must travel a long distance to reach the workplace, as they can save hours by not traveling to the office. Managers have noticed that employees have started to question why they need to come to the office, particularly when many meetings are online - employees are often in online meetings all day long, even when they are at the office. According to number 8 from a public organization:

> *Employees are not going to accept everything—they will be questioning—why do I have to travel to the office? Those who are against traveling to the office have now even more resistance.*

### 3.1.5. Trust

All respondents thought that trust between managers and employees is necessary for telework to be functional. Managers cannot check their employees in the same way as they did before, clarified the interviewees. Telework is often a privilege for employees, and having the option to telework means that the employees have earned the trust of their managers. Number 9 from a public organization said:

> *(As a manager) You have to trust your employees in a different way when they are teleworking. You have to trust them more. You cannot be the controlling manager.*

When managers do not allow their employees to telework, it could be perceived as a lack of trust in the organization. Number 3 from a private organization pointed out:

> *There are still some managers who think that it is good to see the employees at the office most of the time. I think they don't trust people—they are a bit skeptical.*

### *3.2. Task-Based Aspects*
### 3.2.1. Quick Decisions

According to interviewees, telework can lead to quicker and less informed decisions, i.e., employees and managers make their own decisions without discussing the matter with others. This can have negative consequences for the organization in the long term. Number 5 from a public organization described:

> *You will have less information when you make a decision—if you need to discuss it with someone you need to book a meeting. It's not as easy as going to someone and discussing the matter. Sometimes you make quick decisions.*

### 3.2.2. Better Performance

According to the managers, telework makes time management more effective, which is beneficial for the organization. Number 5 from a public organization said:

> *Meetings become more effective online—you go straight to the point. At the office, you speak generally about this and that (before the meeting starts).*

Nevertheless, some meetings are better to have at the office, managers noted—for example, when a working group is brainstorming or developing something together. When managers

want to check the employees' work situation (workload), then it is better to have meetings online. Another advantage of online meetings is that many people can participate in them.

Managers also think that employees' wellbeing and effectiveness have improved due to telework. They are less stressed because they can start to work directly at home without traveling to the office. Employees feel better when they have more flexibility because they can partly choose where and when to work and, therefore, work in a more undisturbed way. Number 16 from a public organization said:

> *Telework gives you a better possibility to work undisturbed. Things that took maybe two hours at the office—you can do now in 45 min when you are working from home.*

Many managers thought that telework and flexibility increase employees' motivation, which improves performance for the whole organization. When employees and managers can choose their work environment, e.g., where they work, it increases their job performance. Number 5 from a public organization pointed out:

> *Flexibility increases motivation . . . Better productivity when you can choose your working environment.*

### 3.2.3. What Do Employees Deliver?

According to the managers, the idea of work has changed due to telework. Work is not related to how many hours an employee works or where—the most important thing is what they deliver. Now, managers pay more attention to the employees' work goals, but not how they achieve their goals. Number 16 from a public organization said:

> *I'm not going to calculate how many working hours they do—but I would notice if they don't deliver or if they are sliding away from the group. They need to take their responsibility to do their work in the best way.*

### 3.3. Added Value

### 3.3.1. Attractivity of the Organization

According to responses, over 90% of the organizations did not regularly use telework before the pandemic. Telework has changed the way in which managers and employees work and, especially, their mindset about telework. All managers mentioned that telework is important for an organization's attractiveness. By offering telework, organizations can attract the competence needed for them to thrive. Therefore, telework can create added value for organizations—it is an important competitive advantage for them. Number 5 from a public organization pointed out:

> *Telework increases the organization's attractivity. If we are going to compete for the workforce, we have to be flexible. How can we become more attractive? Then it is a question about telework.*

Organizations can save time and money in the recruitment process if they offer telework. For some organizations, it has helped them to recruit all kinds of personnel from other cities—they have been able to expand their recruitment areas. For others, telework has facilitated the recruitment of temporary employees, such as consultants. Number 11 from a public organization said:

> *Earlier it was necessary to sit at the office which required that we could only search for consultants who live in the region, otherwise, you need to book a hotel and a business trip, and then it's not sustainable or economical. For us, it has been positive.*

Telework does not only affects recruitment; the managers also thought that if an organization had offered telework and did not continue to offer it, they could lose some employees. Number 5 from a public organization said:

> *If a person thinks it is unnecessary to travel to the office and telework is offered somewhere else, then he or she can change employer. We had one employee who changed his workplace because of a lack of telework.*

### 3.3.2. Traveling Less

Telework has reduced business trips according to the managers. Instead of traveling to clients, managers and employees have meetings online, which is cheaper and more convenient. Number 3 from a private organization said:

> *You don't need to travel to your customers anymore. We can probably reduce traveling (business trips) by 30%.*

## 4. Discussion

The present study aimed to investigate how managers perceived telework arrangements in relation to the work environment and the performance of their organizations. Overall, telework had consequences for both the work environment and the performance of the organizations. Telework mainly affected the organizational and social work environment [26], which included time management—managers and employees could save time in meetings and utilize their effective moments better due to telework. As a result of the occurrence of fewer disruptions during telework, productivity increased. In particular, we found that telework led to (1) changed work routines—some employees preferred more peace and quiet when they worked at the office, and they kept their office doors closed, whereas others preferred social activities; (2) questioning—employees started to question the need to come to the workplace if they were more productive when teleworking; (3) quick decisions—employees and managers made quicker decisions without discussing matters with other colleagues; (4) more delivery-focused work—management paid more attention to what employees delivered than to how they delivered.

### 4.1. Telework—Relationship-Based Aspects

This study found that managers' and employees' work routines had changed due to telework. Some employees and managers wanted to work in their rooms with their doors closed to avoid distractions when they were at the office. Others craved social activities. This could possibly negatively affect creativity and cooperation between work colleagues. Therefore, it is prioritized for management to maintain regular meetings so that working teams do not lose their innovation skills, such as those of critical thinking, complex problem solving, and interaction with colleagues. In fact, this study found that online meetings united the closest working teams. This finding is in agreement with that of a previous study that showed that teleworking may lead to more personal and intense work relationships [27].

The introduction of telework led to employees questioning why they should come to the office at all, especially if they perceived themselves to be more effective when they worked from home. Managers noticed that if they wanted their employees to work more at the office, they needed to have a good reason for it, e.g., discussing working ideas, brainstorming as a team, and strategic planning, which were found to be more effective when work groups met face to face. Then, employees would think that it is worth it to come to the workplace. Lack of motivation to work at the workplace may lead to lower job performance and reassignments (voluntary resignation)—one manager mentioned that at least one employee had quit his job since he was not allowed to telework as much as he wanted. According to Collins et al. [28], if full-time teleworkers needed to return to the office, they were less likely than occasional teleworkers to seek promotion. In addition, teleworkers were generally more satisfied with their jobs than office workers [29]. These findings prompt the question of whether it is the managers' responsibility to motivate employees to come to the workplace. One may argue that employees are being paid to do their job, so it should not be necessary. On the other hand, motivated employees tend to perform better, which is beneficial for the organization [30]. Managers need to find a balance in that regard.

To be functional, telework requires a trusting relationship between managers and employees, as the managers concluded. Regular meetings (online check-ins) are essential, but they should aim to support the employees, not to check their performance. Telework

requires a supportive manager who relies on their employees to do their job. This is in line with previous studies [7,14,15]. Managers who trust their employees and their skills to do their work also tend to perceive telework as a useful work form [15]. If managers check on their teleworking employees due to a lack of trust, it will negatively affect the manager–employee relationship and the whole organization [31]. Nevertheless, in general, managers are concerned that employees lose their integration and bond with the organization when they are teleworking [14,32], which was also discovered in our study. Hence, regular contact between employees and managers, along with support from the manager, may be necessary for making telework functional [31,33].

The fact that telework is not an option for everyone due to work tasks, for example, can cause friction among employees [34]. It can create an "us vs. them" feeling and cause complaining among employees. Some employees may think that it is unfair that others can telework. Overall, around 37% of employees in Europe are in occupations that can technically be carried out from home [35]. This can create inequalities among work colleagues and decrease cooperation among working groups. However, according to Collins et al. [28], when teleworkers regularly visit their offices, it provides a mechanism that helps sustain their relations with office-based colleagues. In addition, a teleworker's reasons for teleworking are crucial for shaping the social relations between teleworkers and their office-based colleagues [27].

### 4.2. Telework—Task-Based Aspects

The managers perceived that telework has affected decision making in their organizations in that employees and managers tend to make quicker decisions at the expense of discussing the matter with other colleagues. While quicker decisions save time, they may not always be beneficial for the organization. According to Nwoye [36], organizations may lose valuable business opportunities if they do not wait for all of the information to be complete before making a decision. However, quick decisions may be needed in a digital era, where time has become an essential factor in business [37], e.g., being first on the market when launching a new product. It also depends on the type of decision—does it affect the whole organization or only one person? Does it have long-term or short-term effects? It is important to find a balance between different kinds of decisions.

The managers reported that telework leads to a larger focus on results, rather than how they are achieved. In other words, an employee can work from anywhere as long as their performance is not affected. This may be a natural consequence of not being able to oversee the work process when employees are teleworking. Possibly related to this perspective, the managers also believed that employees were more effective when teleworking. This is in agreement with previous studies that have shown that telework is positively associated with employee performance and organizational performance [6,12,13,31,32,38]. Further, Kaplan et al. [14] found that working from home was associated with enhanced performance and productivity. Overall, managers are responsible for employees' wellbeing, as employees are an organization's most valuable resource, promoting sustainability in the organization [17]. Consequently, the relationship between a manager and an employee is important for achieving organizational sustainability [17].

### 4.3. Telework—Added Value

Aside from the relationship-based and task-based aspects discussed above, the managers thought that telework could create added value for their organizations. By offering telework, an organization can attract and retain more employees. This makes telework a competitive advantage for organizations. This finding supports those of previous studies that showed that organizations can recruit and retain employees with valuable or rare knowledge and skills [13,39]. There is a positive association between the possibility of teleworking and job attractiveness: 10% more telework hours bring in a rise of 2.2% in job attractiveness [40].

### 4.4. The Perspective of Neo-Institutional Theory

Organizations are closely connected to what goes on in the world around them [41]. They are influenced by societal factors, ideas, rules, fashions, knowledge, ideologies, norms, etc., as well as how these are created and disseminated among organizations [41]. Due to the rapid increase in telework during the COVID-19 pandemic, telework is generally expected to be a regular feature of work, and many organizations need to determine norms and rules for practicing it. In doing so, they need to consider driving forces from within as well as outside the organization.

Among the three institutional pressures of neo-institutional theory, i.e., pressures associated with nations (legislation, policies, and rules), pressures associated with the sector (e.g., competitiveness for labor), and pressures associated with the organization (e.g., proper behaviors according to values and norms in the organization and personal values, beliefs, and assumptions) [18], our findings mostly relate to individual aspects, such as cognitive pressures, e.g., the sub-categories of social community, questioning, work routines, quick decisions, and what employees deliver. Changes in behaviors resulting from telework tend to be driven by individual needs. Perhaps this is not surprising, since telework is often portrayed as a means of balancing work and private life [42].

Examples of sub-categories that fit into the normative pressures are attractive organization and performance. The normative pressure includes values (what is considered proper) and norms enabling institutions to establish ground rules (e.g., some employees are allowed to telework and some are not) [43]. If an organization decides that everyone can telework for 50% of their working time, employees should follow this rule.

Overall, the managers perceived that the use of telework will grow or remain the same. Telework's increased popularity can be partly explained by neo-institutional theory. During the pandemic, the use of telework was strongly recommended by the government in Sweden (pressures associated with nations) [18,43]. After the pandemic, employees wanted to continue to telework—they changed their values related to telework, and it became a personal desire for them, which changed the work habits and norms in organizations. This has affected managers' perceptions of telework—it is more acceptable to telework nowadays than it was earlier, as long as it does not influence employees' work performance.

### 4.5. Methodological Considerations

Rigor is important for demonstrating the legitimacy of the research process [44]. Lincoln and Guba introduced credibility, transferability, and dependability as the major criteria for ensuring rigor in qualitative research (cf. [45]). In this study, these criteria were applied throughout the research process.

This qualitative study aimed to create a deeper understanding of telework among managers in public and private organizations. On the one hand, it is challenging to generalize and replicate the results due to the participants' unique experiences [46]. On the other hand, the flexible nature of the interviews helped the participants reveal more about their experiences, which could enrich the qualitative data [47]. In this study, the sample consisted of male and female managers from different sectors, which made the results transferable to many other organizations in Sweden. The variation in the sample contributed to the transferability of the results to a wider context. However, the majority of the managers were middle managers from public organizations, and a manager's position in the organization and the organizational sector could influence their attitudes towards telework [26].

The interviews were conducted online, and this may have affected our ability to interpret body language and facial expressions. However, the interviewees were familiar with the software used, and we did not consider that it had influenced our results. Credibility and dependability are also questions of how to judge the similarities and variations between categories [45]. In the coding process, all of the authors were involved. The codes were reformulated and changed before the authors decided which code was suitable.

In this study, 90% of the managers and their staff had not teleworked before the pandemic. In other words, they had the short-term experience of managing teleworkers when the interviews were conducted, which may have influenced their perceptions of telework in their organizations. Managers with more experience tended to perceive that the work being done in their areas was compatible with telework, and they were less stressed about it [14].

## 5. Conclusions

The results of this study showed that managers perceived that telework affected both the work environment and the performance of their organizations. Employees changed their behaviors at the workplace; some closed their office doors to avoid distractions, while others valued social activities when working at the workplace. This requires an adaption of the physical work environment to assure that individual needs are accommodated—for example, access to both quiet rooms and areas for creative discussions. This could be one important aspect of making employees want to come to the workplace. The managers also reported that telework could make small close-working teams become even closer, while they could lose contact with others outside the teams. This increases the risk of teams working on their own, causing built-in inefficiency. When organizing work, managers not only have to consider how to build cooperation within teams, but also how to create conditions that enable cooperation among teams in an organization.

In addition, telework led to quicker but less informed decisions, which may benefit productivity in the short term but not in the long term. To facilitate the long-term performance of their organizations, managers have to consider how to organize work to benefit from the increased individual productivity that comes with telework while managing the barriers that lead to making uninformed decisions. Overall, trust between managers and employees was considered crucial for making telework work in an organization. The findings of this study highlight managers' perspectives on the consequences of telework in an organization, which is important for maintaining organizational sustainability. In addition, the results of this study could be used by organizations to create guidelines for telework, which may enhance employees' health and organizational performance in the long run.

*Future Perspectives on Telework*

After the COVID-19 pandemic, telework will likely remain high in organizations that have implemented it. Therefore, research on the long-term effects of telework is needed. This study focused on the organizational perspective, leaving out important aspects such as work–life balance and commuting. Examples of future research areas are the following: How does telework affect employees' health in the long run? Do organizations lose their creativity in the long term when they are focusing more on effectiveness and job performance in the short term?

**Supplementary Materials:** The following supporting information can be downloaded at: https://www.mdpi.com/article/10.3390/su15075845/s1. Supplementary File S1. Interview guide.

**Author Contributions:** Conceptualization: T.K., M.H., K.R. and M.L.-K., methodology: T.K., M.H., K.R. and M.L.-K., software: T.K., validation: T.K., M.H., K.R. and M.L.-K., formal analysis: T.K., investigation: T.K., resources: T.K., M.H., K.R. and M.L.-K., data curation: T.K., writing—original draft preparation: T.K., writing—review and editing: T.K., M.H., K.R. and M.L.-K.; project administration, T.K., M.H., K.R. and M.L.-K. All authors have read and agreed to the published version of the manuscript.

**Funding:** Swedish Research Council for Health, Working life and Welfare—(Forte) with reference number: Ref. No. 2019-01257. The funding body had no influence on the design, process, or conduct of this study.

**Institutional Review Board Statement:** This study was approved by the Swedish Ethical Review Authority (2019-06220).

**Informed Consent Statement:** Informed consent was obtained from all informants involved in this study before the data collection.

**Data Availability Statement:** Supporting data are available upon reasonable request (T.K.; e-mail: tea.korkeakunnas@hig.se).

**Acknowledgments:** We would like to thank the informants for their contributions.

**Conflicts of Interest:** The authors declare no conflict of interest.

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
