# Peer review of "Managers’ Perceptions of Telework in Relation to Work Environment and Performance"

_sustainability, doi:10.3390/su15075845_

Round 1

Reviewer 1 Report

I want to thank the authors for their submission on managers’ perspective of telework. While I think that the paper is well structured and well written, and the applied qualitative approach to data collection and analysis is appropriate, I do not (yet) see any meaningful contribution of this article.

The authors state that “[t]his study aimed to investigate how managers perceive telework arrangements in relation to the work environment and the performance of the organization”. The overall research questions (i.e., the managerial perspective of telework) is not new and has been investigated extensively. The focus on work environment and performance could add something meaningful, but though these aspects are included in the title of the paper and the research question, they do not receive any specific focus in the article.

Therefore, I would ask first of all that the authors situate their contribution more specifically within the extant literature. For an overview they could, for example, consult reviews such as the one conducted recently by Fischer et al. (2021), which includes a host of articles that are also focusing on the managerial perspective of telework, such as:

 ·         Beham, B., Baierl, A., Poelmans, S.: Managerial telework allowance decisions - a vignette study among german managers. The International Journal of Human Resource Management 26(11), 1385{1406 (2015). https://doi.org/10.1080/09585192.2014.934894

·         Brodt, T.L., Verburg, R.M.: Managing mobile work? insights from European practice. New Technology Work and Employment 22(1), 52-65 (2007). https://doi.org/10.1111/j.1468-005X.2007.00183.x

·         Kaplan, S., Engelsted, L., Lei, X., Lockwood, K.: Unpackaging manager mistrust in allowing telework: Comparing and integrating theoretical perspectives. Journal of Business and Psychology 33(3), 365-382 (2018). https://doi.org/10.1007/s10869-017-9498-5

·         Kelly, E.L., Kalev, A.: Managing exible work arrangements in us organizations: Formalized discretion or `a right to ask'. Socio-Economic Review 4(3), 379-416 (2006). https://doi.org/10.1093/ser/mwl001

·         Kwon, M., Jeon, S.H.: Why permit telework? exploring the determinants of California city governments' decisions to permit telework. Public Personnel Management 46(3), 239-262 (2017). https://doi.org/10.1177/0091026017717240

·         Lembrechts, L., Zanoni, P., Verbruggen, M.: The impact of team characteristics on the supervisor's attitude towards telework: A mixed-method study. The International Journal of Human Resource Management 29(21), 3118-3146 (2018). https://doi.org/10.1080/09585192.2016.1255984

·         Neirotti, P., Paolucci, E., Raguseo, E.: Mapping the antecedents of telework diffusion: Firm-level evidence from italy. New Technology, Work and Employment 28(1), 16-36 (2013). https://doi.org/10.1111/ntwe.12001

·         Peters, P., Heusinkveld, S.: Institutional explanations for managers' attitudes towards telehomeworking. Human Relations 63(1), 107-135 (2010). https://doi.org/10.1177/0018726709336025

·         Scholefield, G., Peel, S.: Managers' attitudes to teleworking. New Zealand Journal of Employment Relations 34(3), 1-13 (2009)

·         Silva-C, A., Montoya R, I.A., Valencia A, J.A.: The attitude of managers toward telework, why is it so difficult to adopt it in organizations? Technology in Society 59, 101133 (2019). https://doi.org/10.1016/j.techsoc.2019.04.009

In particular, the authors should then make clear what specifically they contribute to our understanding of the impact of the work environment on the managerial perspective of telework and the relationship with work performance in this regard. Even if this study does not discover anything new, it should at least be clear which aspects that have been reported in previous research did not become apparent in the current study (e.g., maybe social isolation was not as much of an issue as in other studies).

Regarding the applied methodology, I would also ask the authors for some adaptations. First, an overview of the interview participants and some of their main characteristics (e.g., age, gender, job type, type of organization, tenure) would be welcome and in this context, it should then also be mentioned for each interview statement which participant it came from. Second, figure 1 is hopefully just a placeholder, because these types of diagrams are certainly not worthwhile to be published. I would suggest that the authors instead provide their coding structure in a style close or similar to what is recommended by Gioia et al. (2013).

For some minor recommendations, I would also add that the authors should refer to neo-institutional theory instead of simply institutional theory, as this was the basis for the study by Daniels et al (2001). Further, some typographical problems are occurring throughout the paper, which should be addressed. Examples include (non-exhaustive):

Abstract: “…positive and nega-tive consequences…”

p. 12: “…different kinds of decisions.]”

p. 13: “It was grounded on in-terviews with 17 managers…”

p. 13: “…but it could also inte-grate…”

Overall, I see potential in the study, but its likelihood to be published very much depends on the ability of the authors to outline the contribution in relation to the extant body of literature. In any case though, I wish the authors all the best for their future research.

References

Fischer, T., Küll, S., Niederländer, U., and Stabauer, M. 2021. “The New Normal? Motivators for and Hindrances to Telework,” in HCI in Business, Government and Organizations: Proceedings of the 8th International Conference, HCIBGO 2021, Held as Part of the 23rd HCI International Conference, HCII 2021, F. F.-H. Nah and K. Siau (eds.), Cham: Springer International Publishing, pp. 327-346.

Gioia, D. A., Corley, K. G., and Hamilton, A. L. 2013. “Seeking Qualitative Rigor in Inductive Research,” Organizational Research Methods (16:1), pp. 15-31 (doi: 10.1177/1094428112452151).

Author Response

Sincerely,

Tea

Reviewer 2 Report

Dear authors

I would like to thank you for giving me the opportunity to review the manuscript entitled “Managers’ perceptions of telework in relation to the work environment and performance”. The aim of this study was to investigate managers’ perceptions of telework in relation to the work environment and the performance of the organization. As a qualitative study, this study provides valuable information that can be used for future planning to improve the quality of telework and led to better consequences for the organizations. However, I have some suggestions as follows:  

General

- The manuscript should be proofread carefully, there are some mistakes.

Abstract

- More information should be provided regarding methods (such as settings, participant sampling et.).

- The conclusion was not presented in the abstract and you should discuss the messages of the research.  

Introduction

Suitable

Methods

- The interview guide should be provided as a supplementary file.

- What about the rigor of the study? What criteria were used?

Results

- Socio-demographic information of the participants should be provided.

- I think it is better to integrate figures in a single figure to create a more comprehensive image regarding the phenome of the study.  

Discussion

- I suggest summarizing the main findings in the first paragraph of the discussion. 

Author Response

Sincerely,

Tea 

Reviewer 3 Report

This is an interesting paper addressing an important issue that is impacting workplaces around the world.

My comments on specific issues follow.

1.  Please provide discussion of institutional theory in the introduction, as it is well presented in the discussion but some context in the earlier section of the manuscript would be helpful.

2.  How many pilot interviews were conducted and who were the interviewees?

3.  Who conducted the analysis, specifically how many people read the transcripts and were involved in developing the categories presented?

4. A statement in the section on participants regarding the types of organizations represented is needed.  

5. line 231-232 seems to belong below lines 233-238.

6. lines 300-305.  This paragraph is confusing.  The sentences included contradict each other with regard to in-person versus on-line brainstorming.  Can the authors please clarify.

7. 3.3.1 Attractiveness of the workplace-this title is confusing because they authors address the workplace benefits for employees, but the wording implies the visual appeal of the workplace.  Can this be re-worded?

8.  The authors do not really discuss work-life balance issues.  There is no discussion of how disruptive working from home can be for families.  This may be due to the fact that the authors interviewed managers only and that might be a future study worth doing.  This should be mentioned in the discussion.  The other thing that has been discussed in relation to benefits of a commute is that there is time to decompress from stressful work before entering the home and this is lost when telecommuting.

Overall I found the paper well written and interesting.  

Author Response

Sincerely,

Tea

Reviewer 4 Report

Dear authors,

Thank you for sharing your interesting research with Sustainability. Your manuscript requires further editing as commented in the file attached.

Best wishes!

Author Response

Sincerely,

Tea

Round 2

Reviewer 1 Report

I want to thank the authors for their revision. All of my remarks from the previous round have been sufficiently addressed.

Author Response

Sincerely,

Tea

Reviewer 2 Report

Dear authors

Thank you for addressing my comments. However, there are some issues that should be addressed.

1. Please cite the supplementary file (interview guide) in the relevant section of the text. It should be published with the article.

2.  In the QUL study, the researchers should be applied some criteria (such as Lincoln and Guba criteria) to ensure the rigor of the study.  I could not see such an issue in your study.

Best

Author Response

Sincerely,

Tea

Reviewer 4 Report

Dear authors,

Many thanks for your revised manuscript. The manuscript has been very much improved. I suggest you consider the following comments to further enhanced the quality of your manuscript:

1. Having a specific section for the literature review.

2. Elaborate further on the practical contributions of your study.

Thank you and all the best!

Author Response

Sincerely,

Tea
